# An Open-Source Soft Robotic Platform for Autonomous Aerial Manipulation in the Wild

**Erik Bauer**[*], **Marc Blöchlinger**[*], **Pascal Strauch**[*], **Arman Raayatsanati**[*],
**Curdin Cavelti**, **Robert K. Katzschmann**[†]
Soft Robotics Laboratory
Department of Mechanical and Process Engineering
ETH Zurich
{erbauer, mbloechli, pstrauch, araayatsa, cucavelti, rkk}@ethz.ch

**Abstract:** Aerial manipulation combines the versatility and speed of flying platforms with the functional capabilities of mobile manipulation, which presents significant challenges due to the need for precise localization and control. Traditionally, researchers have relied on offboard perception systems, which are limited to expensive and impractical specially equipped indoor environments. In this work, we introduce a novel platform for autonomous aerial manipulation that exclusively utilizes onboard perception systems. Our platform can perform aerial manipulation in various indoor and outdoor environments without depending on external perception systems. Our experimental results demonstrate the platform's ability to autonomously grasp various objects in diverse settings. This advancement significantly improves the scalability and practicality of aerial manipulation applications by eliminating the need for costly tracking solutions. To accelerate future research, we open source[3] our ROS 2 software stack and custom hardware design, making our contributions accessible to the broader research community.

**Keywords:** Aerial Manipulation, Learning-Based Grasping, Autonomous Flight, Robotic Systems, Soft Grasping

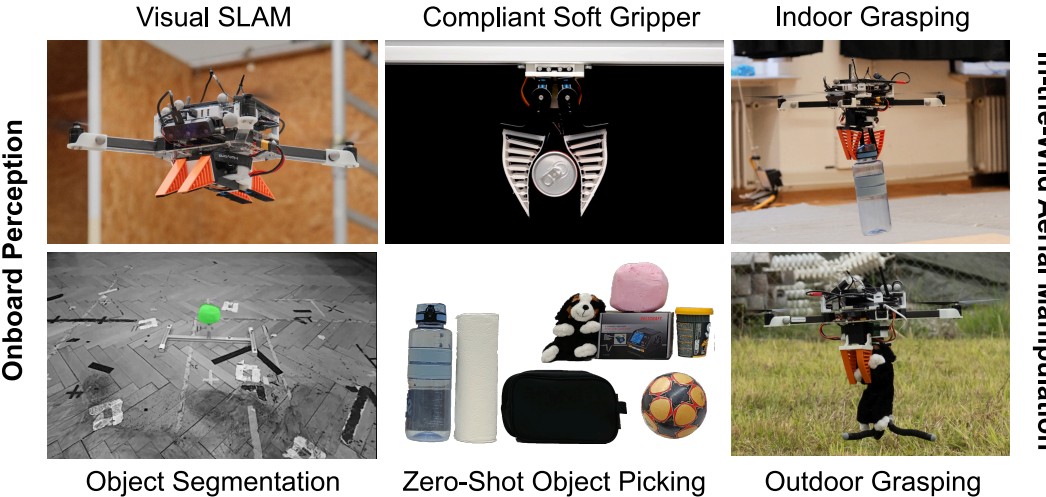

Figure 1: Our autonomous aerial platform demonstrates its aerial manipulation capabilities with onboard-only perception.

---

[*]These authors contributed equally to this work
[†]Corresponding author: rkk@ethz.ch

[3]https://github.com/srl-ethz/osprey

8th Conference on Robot Learning (CoRL 2024), Munich, Germany.

# 1 Introduction

Aerial manipulation, which combines the agility of aerial vehicles with the functional capabilities of manipulators, poses a significant challenge in robotics. Traditional mobile manipulation platforms, typically composed of a mobile base and a manipulator, are designed to interact with their environment and perform various tasks. Aerial manipulators, using multicopters as their mobile base, offer significant advantages over ground-based platforms such as legged robots [1] and wheeled robots [2], particularly in scenarios requiring high traversal speeds or navigation through impassable terrain, such as during floods or in debris-strewn areas.

In recent years, aerial manipulation has seen significant advances using soft robotic manipulators [3, 4, 5, 6]. The compliant properties of soft manipulators relax the otherwise high constraints on precisely positioning the manipulator when grasping objects. Furthermore, soft manipulators also dampen the impact on the dynamics of the flying platform, which eases the burden on flight controllers during each grasp.

However, a significant limitation of existing aerial manipulation platforms is their dependence on external perception systems, such as motion capture systems [4, 5, 7], fiducial markers [3, 8], and VIVE trackers [9], to perform self-localization and target object localization. These systems simplify the research problem by providing nearly perfect pose information, but also present two core issues.

First, external perception systems, particularly for self-localization, assume access to almost flawless pose data. Such strong assumptions often limit the applicability of research findings, increasing the barrier to adaptation in real-world scenarios. Second, motion capture systems can be prohibitively expensive, creating a high financial barrier that restricts initial research efforts and limits the reproducibility of existing studies.

To overcome these limitations, we propose a new platform that relies exclusively on onboard perception sensors for aerial manipulation. For self-localization, we use an existing simultaneous localization and mapping (SLAM) system [10], allowing the platform to estimate its state using visual observations alone. We implement a learning-based pipeline to localize target objects through RGB-D segmentation and point cloud analysis to determine target grasp locations.

Our approach enables autonomous aerial outdoor manipulation without requiring significant prior knowledge about the environment or target objects. This work introduces a novel method that relies solely on onboard perception, distinguishing itself from previous approaches by operating without strong object priors such as computer-aided design (CAD) models or semantic object categories. To our knowledge, this represents the first demonstration of such capability in autonomous aerial outdoor manipulation, allowing the platform to adapt to diverse and unpredictable scenarios. With our platform, we aim to facilitate future research in key areas of aerial manipulation, including hardware optimization, perception, planning, and control, by providing an open-source, modular framework that researchers can easily build upon and adapt.

## 1.1 Contributions

Our contributions can be summarized as follows:

1. We propose an autonomous soft robotic aerial manipulation platform utilizing soft robotics, SLAM for self-localization, and learning-based target localization. This platform demonstrates zero-shot manipulation capabilities for diverse objects across indoor and outdoor environments.

2. We open source our ROS 2 software stack, our custom onboard power electronics stack, and our custom hardware design to facilitate further research on autonomous aerial manipulation.

3. We validate the platform's effectiveness and versatility through flight and grasping experiments, demonstrating its ability to autonomously grasp a number of objects with a high success rate.

## 2 Related Work

The integration of aerial systems with grasping mechanisms has gained considerable attention in recent years [11]. Research has demonstrated the potential of aerial grasping using soft grippers, which adapt to the shapes of objects due to their passive compliance [3]. Further advances include rapid grasping platforms [4] and efforts to achieve more autonomous grasping without relying on fiducial markers [12].

However, many existing systems still depend on motion capture systems to provide the precise location of the aerial platform or the object. For example, several works have focused on improving flying manipulators with stiffness control to improve object interaction dynamics [13] and developing lightweight, low activation force grippers designed specifically for aerial grasping applications [14]. In addition, Wu et. al. [15] has investigated optimizing quadcopter designs to enhance aerial grasping performance.

Despite these advancements, the reliance on motion capture systems for self-localization and object priors for target localization remains a common theme. More recent work has demonstrated successful aggressive aerial grasping using onboard perception by employing visual-inertial odometry for self-localization and CAD models for object recognition [5]. Our approach differs from this by leveraging a learning-based target localization system for zero-shot video segmentation integrated with a SLAM pipeline. This method eliminates the need for prior assumptions regarding object geometry or the environment, enhancing the flexibility and applicability of aerial grasping systems in unstructured settings.

## 3 System Architecture

In this section, we motivate the key design choices that form the architecture of our proposed aerial system. Due to the highly specialized nature of aerial grasping, we have designed a custom modular quadrotor frame and power electronics stack to facilitate platform development. Additionally, we have developed a custom software stack for seamless and modular development on ROS 2 in conjunction with the PX4 flight-control stack. We open source our custom hard- and software as an accessible starting point for other researchers in the field.

### 3.1 Hardware Stack

#### 3.1.1 Platform

The platform is designed with a focus on modularity and maintainability. Compared to consumer-grade "ready-to-fly" platforms, which are commonly modified to serve as research platforms, our design allows for simpler customization to accommodate various downstream applications, including aerial manipulation.

The platform has an x-shaped configuration optimized for minimized and symmetric inertia in roll and pitch. Our multi-level design requires a vertical assembly: gripper components are located at the bottom, electronic circuit boards and flight controllers occupy the middle level, and the onboard computer and gripper controller are mounted above. This layout simplifies access to components that require frequent adjustments. Acrylic plates provide visual access while protecting the internal components. Electronic speed controllers (ESCs) are integrated within carbon profile arms, and frame connections are made of 3D-printed polylactic acid. The takeoff weight of the platform, including the battery, is $2479\,\mathrm{g}$, resulting in a payload capacity of approximately $500\,\mathrm{g}$.

A custom-developed power management board offers overcurrent protection, weighs less, and is more compact than off-the-shelf components. The gripper features two palms, both with two fingers inspired by the Fin Ray® effect [16]. The fingers leverage their compliance to bend around target objects upon contact, creating reliable grasps regardless of object geometry. The fingers are 3D-printed with 85A thermoplastic polyurethane to offer the necessary flexibility and strength for effective manipulation.

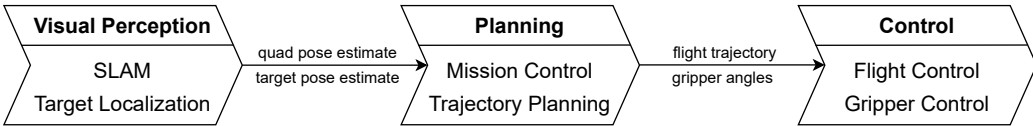

Figure 2: The overall pipeline of our aerial manipulation system. The visual perception module includes SLAM for estimating the system's pose and target localization to identify objects. The planning module encompasses mission control and trajectory planning based on visual estimates. Finally, the actuation module involves flight and gripper control to execute the planned trajectory and manipulate objects.

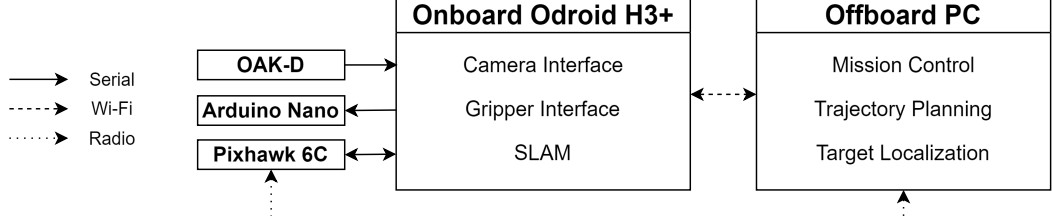

Figure 3: The component architecture with corresponding connections. The onboard components, including the OAK-D, Arduino Nano, and Pixhawk 6C, are connected to the Odroid H3+ via USB connections. The onboard and offboard computers maintain a WiFi connection for exchanging data between the modules. The offboard PC also establishes a radio connection to the Pixhawk 6C flight controller, allowing the offboard modules to communicate directly.

## 3.2 Middleware

Our software design follows the typical autonomy pipeline, including perception, planning, and control routines. The logic flow is illustrated in Figure 2, and the architecture of the software components is shown in Figure 3. For low-latency communication, we use ROS 2 [17] instead of ROS 1 [18], which is still the predominant middleware in most current works [3, 7, 13, 19, 20]. Unlike ROS 1, ROS 2 supports real-time communication, which is highly advantageous for applications requiring aggressive flight maneuvers [3, 4, 5].

## 3.3 Planning, Perception, Control

Our visual perception system uses an OAK-D S2 camera, which provides aligned grayscale and depth images at 30 Hz to both the SLAM pipeline and the target localization module. The SLAM pipeline also incorporates IMU data streamed at 200 Hz. Using a single camera for both SLAM and downstream perception, we reduce overall weight and complexity compared to setups that use separate cameras for visual-inertial odometry (VIO) or SLAM and downstream perception [5].

For planning, a mission control module manages the overall mission logic, observes the current state, and detects and handles exception states or failure modes. Working closely with this module, a trajectory planner computes the subsequent flight maneuvers based on the current mission state and specifies reference trajectories to the Pixhawk flight controller. The Pixhawk 6C runs PX4 Autopilot [21], an open-source flight control system that communicates over MAVLink [22]. To control the gripper in parallel, a separate ROS 2 interface receives commands over the network and forwards them to an Arduino Nano via UART serial communication.

During testing, we frequently employ a VICON motion capture system to obtain ground-truth pose information on the quadcopter and the object. These ground-truth estimates are also used to test various modules in isolation. To access the VICON system measurements and publish them to the ROS 2 network, we employ an open-source repository from OPT4SMART [23]. The VICON system is configured to provide measurements at 200 Hz.

### 3.4 Self-Localization

Achieving robust motion estimation through visual observations remains a challenging problem. While different open-source methods exist that perform well in synthetic benchmarks [24, 25, 26, 27, 28], we find that most of these published methods suffer from unsatisfactory robustness across different real-world scenarios. Multiple problems arise: high sensitivity to the camera and IMU calibration values makes it challenging to get repeatable localization results due to minor differences in calibration. Other parameters are often fine-tuned for specific synthetic benchmarks, necessitating a lengthy search to make pipelines work across different real-world environments.

The modular design of our pipeline allows for seamless integration of various SLAM algorithms. We conducted a comprehensive comparison of several open-source options, including SVO [28], Kimera [27], Rovio [24], and RTAB-Map [29]. Our evaluation revealed that Spectacular AI [10] consistently provided the most robust and precise state estimates. Notably, the modular nature of our system enables researchers to easily substitute any alternative SLAM pipelines as baselines when developing new state estimation algorithms. Further comparison details can be found in the Appendix (Section 6.3).

### 3.5 Target Localization

We propose a novel target localization pipeline, which can be used for zero-shot grasping of different objects. In contrast to existing pipelines for aerial manipulation [5, 12], this approach requires neither CAD models nor a semantic class label of the target objects, which enables strong generalization for zero-shot grasping.

The pipeline is visualized in Figure 4. The process is initialized with input from the operator, who selects a point of the target object in a livestream of the onboard camera. To produce a segmentation mask across subsequent image frames, we use SAM2 [30]. Given the segmentation mask and its corresponding depth image, we obtain a partial point cloud of the object.

We select the grasping points [31] using the following strategy. First, we estimate the centroid of the point cloud and determine its base. We then duplicate the point cloud and rotate it by 180 degrees (assuming symmetry of the object) to estimate its front and back. Using this assumed symmetry, we determine a new centroid for the object. Finally, we determine a set of candidate points for grasping. We do so by taking the intersection of the point cloud and a cutting plane that goes through the estimated centroid and is normal to the longest axis.

## 4 Experiments

To evaluate the capabilities of our proposed system, we conduct two experiments. First, we compare the accuracy of the SLAM system to that of a motion capture ground truth to quantitatively determine the quality of the state estimates using only onboard perception. Second, we show a comprehensive overview of grasp success rates for 8 commonly used household objects, using only onboard perception to perform all grasps without any explicit prior knowledge of these objects.

### 4.1 Flight Experiments

First, we aim to investigate the pose errors induced by the Spectacular AI SLAM pipeline. To compute the relative pose error (RPE), we assume that the quadcopter pose in the motion capture system is the ground truth. In Figure 5, we visualize an exemplary grasping trajectory expressed in the north-east-down convention, along with ground-truth data from a motion capture system. The error appears to correlate with the platform's velocity, observing a peak RPE of 0.042 during a fast descent toward the object.

Overall, real-world tracking performance meets the high-precision requirements of aerial manipulation tasks as long as it initializes well. To ensure sufficient image features are present in the camera

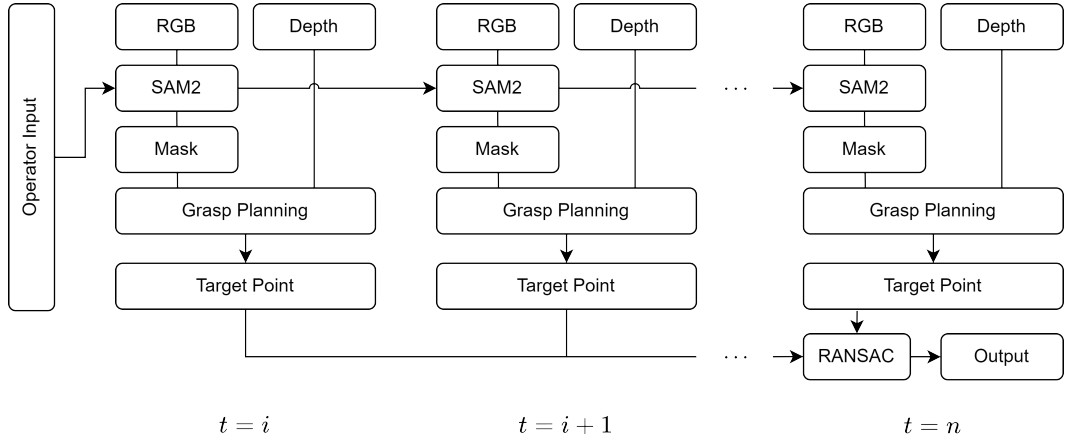

$t = i$          $t = i + 1$          $t = n$

Figure 4: A high-level overview of the target localization pipeline. The target localization process starts at timestep $t = i$, with the operator selecting the target object in an RGB livestream of the onboard camera. We use SAM2 to segment and track the target object across subsequent frames. Each mask is fused with the corresponding depth image to produce a partial point cloud of the object, which is then used for grasp planning. The individual target point estimates are fused via 1-point RANSAC to compute a robust estimate of the target pose.

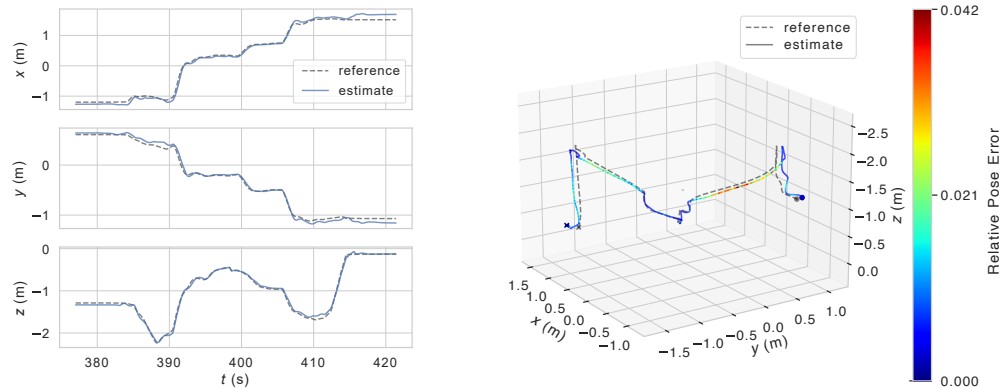

(a) Reference (motion capture) and estimate (SLAM) translation

(b) Relative pose error (RPE) in between the reference and estimate trajectory

Figure 5: We evaluate the accuracy of the Spectacular AI SLAM system against motion capture ground truth during a sample grasping mission. All coordinates are expressed in NED (north-east-down) convention. The two trajectories have been aligned using Umeyama alignment with EVO [32]. We observe a peak RPE of 0.042 during a fast descent towards the object, whereas the mean RPE is 0.028.

frames at SLAM initialization, we find that increasing the platform's initial position height to at least 80 cm above ground significantly boosts the quality of SLAM initialization.

## 4.2 Grasping Experiments

To demonstrate the autonomous capabilities of our platform, we performed nine indoor experiments in a controlled setting to ensure repeatability. The general setup consists of a predefined search area with a target object arbitrarily placed within. The platform must first search for, grasp the object within the search area, and end the mission by dropping the object at a predefined location. We employ a simple search algorithm, visualized in Figure 6.

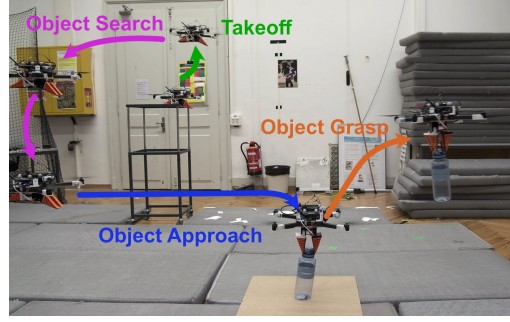

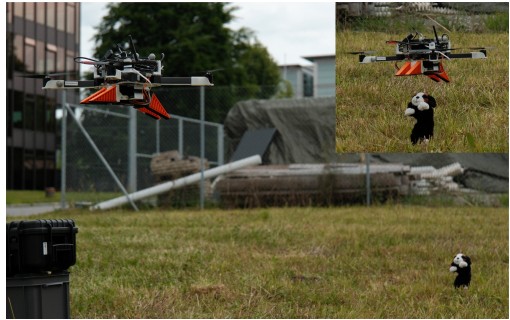

(a) Illustrated standard mission for searching and grasping an object.

(b) Same mission as in a), but doing it outdoors with a plush toy instead of a bottle.

Figure 6: The platform executes the same mission indoors and outdoors. First, the platform iteratively advances through a predefined search field. Upon receiving input from our segmentation system (section 3.5, the platform approaches the target object to refine the pose estimate. Based on the refined estimate, it performs a grasping maneuver.

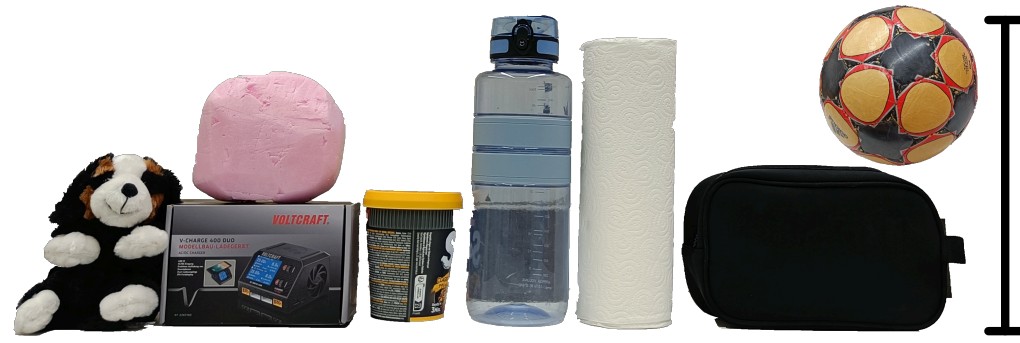

Figure 7: The eight objects we use for grasping experiments shown in Table 1. From left to right: plush toy, styrofoam object, cardboard box, ramen cup, bottle, kitchen roll, pouch, ball. The objects differ distinctly in size, weight, shape and surface friction. The bar denotes a length of $30\,cm$.

## 4.3 Results

In Table 1, we present the success rates for nine experiments obtained in the indoor setting using onboard perception only. Across a total of 144 grasp attempts, our system exhibits an average grasping success rate of 85%, which falls in line with previous works that rely on external perception systems [3, 4, 12]. In contrast to these works, we show zero-shot generalization to different objects by eliminating the need for explicit object priors in our grasp planning algorithm.

We can observe different failure modes. The most common failure modes are horizontal position errors incurred by the increasingly difficult control as the platform approaches the ground. These position errors can then cause the gripper to miss the target object. Secondly, we can observe that for objects with low height, the platform rarely closes the gripper above the object, which could be attributed to increased thrust closer to the ground.

Furthermore, we find that the ability to grasp objects autonomously transfers to other environments, such as outdoor settings, as shown in Figure 6, given that there are no significant disturbances that impact the controllability of the platform (e.g. wind gusts).

Overall, there appear to be few to no downsides concerning the efficacy of autonomous grasping using onboard perception. In contrast, the upsides of exclusively relying on onboard perception constitute a significant step closer to scalable real-world applications of aerial manipulators across different scenarios, allowing for reliable flight without installing extensive infrastructure.

| Object | Success Rate | Attempts | Weight [g] | Dimensions [cm] |
|---|---|---|---|---|
| Bottle (upright) | 75% | 16 | 248.9 | 10 x 10 x 29 |
| Bottle (sideways) | 81% | 16 | 248.9 | 10 x 29 x 10 |
| Plush Toy | 75% | 16 | 68.8 | 10 x 12 x 26 |
| Pouch | 94% | 16 | 240.9 | 8 x 25 x 15 |
| Styrofoam Object | 94% | 16 | 88.7 | 9 x 15 x 13 |
| Kitchen Roll | 81% | 16 | 140.1 | 9 x 9 x 27 |
| Ramen Cup | 81% | 16 | 120.4 | 10 x 10 x 12 |
| Ball | 88% | 16 | 143.8 | 14 x 14 x 14 |
| Cardboard Box | 100% | 16 | 93.0 | 8 x 17 x 11 |
| Total | 85% | 144 | | |

Table 1: Success rates, number of attempted grasps, weight, and dimensions for test objects in the indoor setup to ensure repeatable tests.

## 4.4 Limitations

We occasionally encountered issues with initializing the platform's starting pose, leading to unexpected rotations at takeoff. This resulted in a misalignment between the actual search pattern and the expected search sector, potentially causing the platform to miss the target object. To mitigate this problem, we placed the platform on an elevated surface to enhance SLAM initialization.

Due to the gripper's placement below the quadcopter, all grasped objects are subjected to significant aerodynamic downwash. Consequently, the object's position may shift during the grasping attempt when it is out of the camera's field of view. To minimize this issue, most grasping objects were placed on surfaces with artificially increased friction or mounted on stable supports (e.g. duct tape).

Our grasp planning algorithm operates on partial object point clouds and assumes approximate symmetry of target objects. This assumption limits the grasp planning capabilities of the platform, which could be addressed using point cloud reconstruction techniques [33] Furthermore, long-term video segmentation and tracking is not possible with SAM2, as the amount of memory and compute time increases with each frame. While sufficient for short grasping missions, using only short-term video segmentation (less than 400 frames) could be a limitation for other usages.

## 5 Conclusion

In this work, we have introduced a new autonomous robotic aerial manipulation platform that can localize and grasp targets indoors and outdoors. Using an off-the-shelf SLAM system for self-localization and a learning-based pipeline for target object localization, our platform can operate independently of external perception systems. This autonomy removes the constraints imposed by external motion capture systems, enabling the platform to perform aerial manipulation in various environments, including challenging outdoor settings. With 144 flights for nine zero-shot grasping experiments, we demonstrate that our platform shows competitive grasping efficacy (85%) compared to previous state-of-the-art works using external perception systems.

Our approach addresses significant limitations in the current state of aerial manipulation research, particularly the reliance on pose information provided by external systems and the high costs and limited mobility associated with such equipment. By eliminating these dependencies, our platform broadens the applicability of aerial manipulation. In addition, the open-source release of our ROS 2 software stack, onboard power electronics, and custom hardware design aims to lower the initial investments to facilitate research and development in autonomous aerial manipulation with soft robots.

In summary, our contributions demonstrate the feasibility of autonomous aerial manipulation using onboard perception and pave the way for future research and real-world applications in this domain. Future work could address non-symmetric objects in grasp planning and downwash effects.

**Acknowledgments**

We would like to thank Aashi Kalra for her valuable support in leading the design of the power management board, its assembly, and testing. She has been a valuable member of the team and the project. We also express our gratitude to Jero Barahona, who assisted the design of the power management board.

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

# 6 Appendix

## 6.1 Parts List

Additional resources for replicating our hardware platform are available in our public repository at https://github.com/srl-ethz/osprey. All files for sketches, 3D-printed linkages, laser-cut plates and an assembly guide can be found there. Given an RC Control unit, a workshop, and miscellaneous small parts such as screws, the total cost should be around $ 1274. See Table 2 for the item list.

| Count | Name | Price per Unit |
|:---:|:---|:---|
| 1 | Odroid H3+ | $ 165 |
| 1 | NVME SSD | $ 50 |
| 2 | 16GB DDR4 RAM | $ 49 |
| 1 | OAK-D | $ 250 |
| 1 | Pixhawk 6C | $ 140 |
| 4 | Motor 2216-920KV | $ 20 |
| 4 | Propeller 1045 | $ 6 |
| 4 | 20A ESC | $ 14 |
| 1 | LiPo 4300 mAh | $ 80 |
| 1 | Arduino Nano | $ 20 |
| 1 | Telemetry Transmitter | $ 60 |
| 1 | Custom PCB | $ 40 |
| 1 | 3D printed parts | $ 30 |
| 4 | Carbon Arms | $ 50 |
| 3 | Acrylic Plates | $ 10 |
| **Total Price** | | **$ 1274** |

Table 2: Total Price of the Platform

## 6.2 Power Management Board

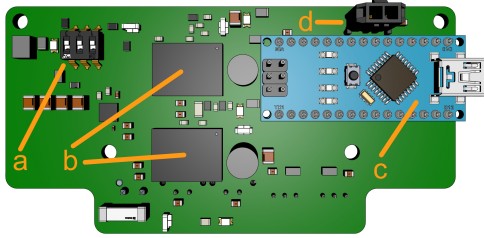 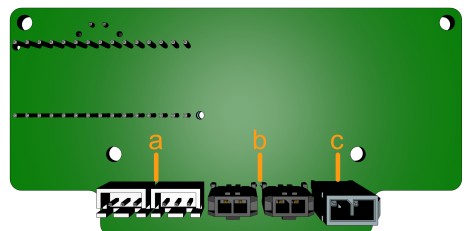

(a) Top side of the custom PMB: a) on-off switches for power circuits b) DC/DC converters c) Arduino Nano d) power connector to onboard PC

(b) Bottom side of the custom PMB: a) servo connectors b) 5 V connectors c) lithium polymer battery input

Figure 8: 3D visualization of the custom power management board (PMB), designed to centralize and regulate power distribution for multiple voltage requirements across the platform.

Our custom PMB, shown in Figure 8, supplies power to the onboard computer (19 V at up to 2 A), the gripper's servo motors (6 V at up to 3 A), and various onboard electronics (5 V at up to 2 A). The Arduino Nano is directly connected to the gripper's servos to relay setpoints from the onboard computer.

### 6.3 Comparison of SLAM Algorithms

In the following, we go over different SLAM algorithms which we tested on our onboard computer. To support the use of ROS 1, we utilize the *ros1_bridge* package. We used each SLAM system with the provided default parameter set, adjusting it as needed to a reasonable degree. To this end, this overview is not exhaustive and further performance gains could likely be obtained by spending more time tuning the different algorithms. An in-depth quantitative overview and comparison of different SLAM algorithms can be explored in future work with a stronger focus on self-localization.

- SVO Pro [28]: very sensitive to camera calibration. Very little documentation. Compatible with ROS 1.

- Kimera [27]: precise for slow maneuvers, but does not run fast enough for faster maneuvers without sacrificing accuracy. Compatible with ROS 1.

- Rovio [24]: similar to Kimera, does not run fast enough to fly agile maneuvers without sacrificing accuracy. Compatible with ROS 1.

- RTAB-MAP [29]: similar to Kimera, does not run fast, relocalization capabilities are not robust enough for agile drone flight. Compatible with ROS 1.

- Spectacular AI [10]: offers out-of-the-box SLAM, which runs fast and offers good precision. Template wrappers are provided for ROS 1 and ROS 2. Closed-source core algorithm.

Other common tracking solutions include the Intel RealSense T265 camera, which offers out-of-the-box tracking, but is no longer produced and suffers from poor availability. Another off-the-shelf alternative is the VIO pipeline provided by ZED. However, the ZED SDK requires CUDA on the host computer, which greatly constrains the choice of onboard computers because it requires a CUDA-compatible graphical processing unit.

