# OpenReview forum: "An Open-Source Soft Robotic Platform for Autonomous Aerial Manipulation in the Wild"
_robot-learning.org/CoRL/2024/Conference — CoRL 2024_

### Official Review · Reviewer_BFE8 · 2024-07-19
**An interesting work and kudos to the opensource effort. But seemingly lack algorithmic contributions... Decision updated on Aug 15**

**Originality:** 2
**Technical Quality:** 4
**Clarity Of Presentation:** 4
**Potential Impact:** 3
**Recommendation:** 3
**Confidence:** 4

**Review:**

The paper clearly presented the open-sourced platform. Based on the demo video in the attachment, I think the platform is suitable for benefiting other research activities that can help improve the status quo of aerial manipulation.

My concern is that the contributions seem inadequate in this work because it is hard to tell what unique contributions are made by the authors in this platform that are not taken from an existing hardware/software/algorithm. Maybe the power module board is new, but the novelty is comparatively minor. Besides, when the authors claim that this is the first work of autonomous aerial outdoor manipulation using onboard perception, they should be aware of the following literature that has achieved it already:

Ubellacker, Samuel, Aaron Ray, James Bern, Jared Strader, and Luca Carlone. "Aggressive Aerial Grasping using a Soft Drone with Onboard Perception." arXiv preprint arXiv:2308.06351 (2023).

One suggestion to the authors is to really develop a new learning-based algorithm in the task of object manipulation given the convenience of experimentation enabled by this platform. I appreciate the efforts of the authors to opensource this architecture. But the contributions do not match the bar of this conference.

update on Aug 15: the decision has been changed.

**Quality Of The Limitations Section:**

1

**Questions For Rebuttal:**

None.

**Robotics Focus:**

4

**Summary Of Paper:**

This paper introduces an opensource quadrotor platform that carries a soft gripper for aerial manipulation. The platform carries stereo cameras for localization and target detection. The entire autonomy pipeline is introduces as well as the hardware components onboard. I have a hard time summarizing the contributions in this works because I think the major efforts were on the integration existing solutions to build this platform. No specific learning-relevant algorithms or designs have been discussed.

**Summary Of Recommendation:**

I recommend rejecting this paper because of lack of sufficient contributions.

---

### Official Review · Reviewer_KZo7 · 2024-07-21
**The paper proposes an aerial manipulation framework for drones equipped with a soft robotic gripper to fetch objects and transfer them from one point to another.**

**Originality:** 4
**Technical Quality:** 4
**Clarity Of Presentation:** 5
**Potential Impact:** 4
**Recommendation:** 3
**Confidence:** 4

**Review:**

The paper is well written, clearly organized as well as well motivated. The hardware design is carefully curated in order to cater to the problem. Since the project is also open-sourced, it will be an impactful contribution to build upon. While the task of aerial manipulation is particularly challenging yet impressive, the results both qualitative as well as quantitative show promise. It would have been interesting to see the experiments on a more diverse variety of objects where the adaptability of a soft-gripper can be clearly seen such as oddly shaped objects, or objects of different shapes and sizes. Certain key factors such as reliance on symmetry of the object (as per the pre-processing of the partial point clouds in a certain way) and laser sensor for when the object is not in the field of view might be a limiting factor for its generalizability. Localization and stability challenges due to sensor noise as well as mobile manipulation pose additional challenges for the successful deployment of such a system.

**Quality Of The Limitations Section:**

3

**Questions For Rebuttal:**

I request the authors to address the following concerns:

1. It is unclear how the agent deals with the uncertainty estimates from the time of detection when the object is in the field-of-view of the onboard camera to actually grasping it when the camera cannot see it anymore. How does it affect based on the size of the object and hence the precision required to successfully grasp it?

2. Does the agent behavior change based on the changes in the physical properties of the object such as its mass, odd/asymmetric shape as well as the coefficient of friction?

3. How does poor depth under inappropriate lighting conditions or holes in the depth image under near proximity affect the visual sensing of the agent?

**Robotics Focus:**

4

**Summary Of Paper:**

This paper proposes an interesting aerial manipulation problem setup, tested in the wild using an onboard perception sensor. While traditional methods rely on off-board vision such as using motion capture, this work uses an onboard sensor, allowing it to efficiently perform the task in both indoor and outdoor settings in a cheap and effective way without relying on expensive tracking solutions.

**Summary Of Recommendation:**

This paper presents some novel and interesting results in the field of aerial manipulation along with hardware experiments. The technical contributions are significant, however the experimental results could have been more diverse and exhaustive, displaying the compelling efficacy of the proposed approach. I propose to give it a weak acceptance.

---

### Official Review · Reviewer_ouWQ · 2024-07-22

**Originality:** 3
**Technical Quality:** 2
**Clarity Of Presentation:** 3
**Potential Impact:** 3
**Recommendation:** 3
**Confidence:** 3

**Review:**

The paper presents an interesting alternative to mobile manipulation for moving objects around which I have not seen before - mounting a gripper on a drone and using it to grab objects. However, I think the platform is still not at the stage where it can be published and used by the robotics community.
- The paper claims that the platform is completely open source and unlike prior work only relies on onboard perception. However, the key to enabling this is a proprietary SLAM algorithm from Spectacular AI. Using proprietary SLAM makes it impossible for researchers to modify and improve the onboard perception of the robot.
-  I am unsure of the utility of this platform because of the limited evaluation. Results are only shown on two objects - a water bottle and a plushy. The former is only shown in a single upright pose, while the latter is highly deformable and very forgiving to incorrect grasps. It is unclear whether their platform would work for more complex objects.
- The authors are proposing a platform for research, but it is not discussed what research problems it can be used to study. When proposing a platform, ideally authors should showcase multiple potential areas for research and how their platform can help. For example, they could propose a benchmark for control / perception or agile grasping with existing baselines, train / test splits. For eg, they can provide an open source simulator and sim2real pipeline for researchers to study sim2real techniques. In its current form, it is unlikely anyone would adopt this platform.

**Quality Of The Limitations Section:**

3

**Questions For Rebuttal:**

- When conducting grasping experiments, to what degree is the initial pose of the object varied? Eg. what happens if the bottle is kept on its side instead of upright?
- What are the main failure modes observed in the case of the plushy and the bottle?

**Robotics Focus:**

4

**Summary Of Paper:**

This paper presents an open source system for aerial manipulation research that relies solely on onboard sensors for perception. This includes the choice of quadruped, OAKD camera, laser range finder, choice of SLAM algorithm, object detection stack, and hardcoded grasping routine. The authors demonstrate grasping results on two objects - a plushy and a water bottle.

**Summary Of Recommendation:**

I vote to weak reject this paper because in my opinion the proposed platform is not yet ready for use by the wider robotics community due to its reliance on proprietary SLAM + limited demo results.

---

### Author Rebuttal · Authors · 2024-08-12

We sincerely thank the meta-reviewer and all reviewers for their thoughtful feedback. We've addressed the concerns raised and significantly improved our manuscript:
1. Research Questions and Use Cases: We've expanded Section 1 to highlight key research directions our platform enables:

- Hardware design optimization for aerial manipulation
- SLAM/odometry algorithms for aerial contexts
- Perception techniques for aerial grasp planning
- Motion planning in cluttered 3D environments
- Grasp trajectory optimization

Our modular framework allows researchers to focus on specific components without implementing the entire system, lowering the entry barrier to this field.

2. Experimental Diversity: We've conducted extensive additional testing:

- Updated grasp planning pipeline using a class-agnostic segmentation model (SAM2) for zero-shot grasping
- Demonstrated grasping of 8 different objects across 144 trials, achieving 85% average success rate

This expansion showcases our platform's robustness and versatility.

3. Novelty and Contributions: We've clarified our novel contributions:

- First open-source platform for autonomous outdoor aerial manipulation without prior object knowledge
- Distinguished by reproducibility (all software/hardware open-sourced) and zero-shot grasping capabilities

4. Specific Responses:

- SLAM: We've tested various algorithms (SVO Pro, Kimera, Rovio, RTAB-Map) but chose a proprietary solution for robust performance given limited onboard compute. Our framework supports the easy integration of custom SLAM algorithms.
- Failure modes: Added discussion in Section 4.3. Main issues are horizontal position errors near ground and gripper misses for low-height objects.
- Object pose variation: New experiments include various object poses, e.g., 81% success for sideways bottle vs. 75% upright.


5. Conclusion

We believe these improvements and clarifications significantly strengthen our paper. By expanding our experiments, clarifying our novel contributions, and addressing specific technical concerns, we have directly responded to the weaknesses identified while building upon the strengths recognized by the reviewers. Our work now presents a more comprehensive and robust contribution to the field of aerial manipulation, offering a versatile, open-source platform that we believe will accelerate research in this exciting area.
We thank the reviewers for their valuable input and hope that these revisions adequately address their concerns.

---

### Decision · Program_Chairs · 2024-09-04

**Decision:**

Accept

**Comment:**

The authors propose an open platform for robotic manipulation. The system is unique, and will hopefully enable more future work. The authors showed that the system is versatile and produce a system that should be reproducible, and can grasp a wide variety of different objects. They describe many different potential research directions enabled by their device, and go to lengths to make sure it is reproducible. The paper is clear and reasonably well-written.

Strengths:
- Grabbing objects with a gripper on a flying drone is interesting and a challenging problem
- System seems very novel and proposes a unique problem
- The system is mostly open-source and seems like it will be reproducible by other labs

Weaknesses:
- Authors should better describe what research questions they think the platform can be used for
- Experiments don't show it being used on very many objects
- Authors should clarify their novelty and contributions